# Real-World Efficacy and Safety of First-Line Nivolumab Plus Chemotherapy in Patients with Advanced Gastric, Gastroesophageal Junction, and Esophageal Adenocarcinoma: A Nationwide Observational Turkish Oncology Group (TOG) Study

**DOI:** 10.3390/cancers16122251

**Published:** 2024-06-18

**Authors:** Yasin Kutlu, Shute Ailia Dae, Feride Yilmaz, Dilek Erdem, Mehmet Ali Nahit Sendur, Sinem Akbas, Elif Senocak Tasci, Onur Bas, Faysal Dane, Abdullah Sakin, Ali Osman Kaya, Musa Baris Aykan, Yakup Ergun, Sedat Biter, Umut Disel, Mustafa Korkmaz, Fatih Selcukbiricik, Fatih Kose, Omer Fatih Olmez, Ahmet Bilici, Gokhan Demir, Suayib Yalcin

**Affiliations:** 1Department of Medical Oncology, Faculty of Medicine, Medipol University, Istanbul 34083, Turkey; 2Department of Medical Oncology, Faculty of Medicine, Baskent University, Adana 01140, Turkey; 3Department of Medical Oncology, Faculty of Medicine, Hacettepe University, Ankara 06800, Turkey; 4Department of Medical Oncology, VM Medical Park Samsun Hospital, Samsun 55200, Turkey; 5Department of Medical Oncology, Ankara Bilkent City Hospital, Ankara 06800, Turkey; 6Department of Medical Oncology, Faculty of Medicine, Koc University, Istanbul 34460, Turkey; 7Department of Medical Oncology, Acibadem Atakent Hospital, Istanbul 34303, Turkey; 8Department of Medical Oncology, Acibadem Altunizade Hospital, Istanbul 34662, Turkey; 9Department of Medical Oncology, Medipol University Bahcelievler Hospital, Istanbul 34196, Turkey; 10Department of Medical Oncology, Medicana International Hospital, Istanbul 34520, Turkey; 11Department of Medical Oncology, Gulhane Training and Research Hospital, Ankara 06010, Turkey; 12Department of Medical Oncology, Antalya City Hospital, Antalya 07200, Turkey; 13Department of Medical Oncology, Faculty of Medicine, Cukurova University, Adana 01330, Turkey; 14Department of Medical Oncology, Acibadem Adana Hospital, Adana 01130, Turkey; 15Department of Medical Oncology, Tokat State Hospital, Tokat 60100, Turkey; 16Department of Medical Oncology, Acibadem Maslak Hospital, Istanbul 34398, Turkey

**Keywords:** nivolumab, chemotherapy, gastric adenocarcinoma, real-world, efficacy, safety

## Abstract

**Simple Summary:**

We evaluated the real-world efficacy and safety of nivolumab plus chemotherapy in patients with HER2-negative unresectable advanced or metastatic gastric, gastroesophageal junction (GEJ), or esophageal adenocarcinoma. In addition, we identified subgroups that may experience superior outcomes. The median progression-free survival and overall survival were 11.7 months and 18.2 months, respectively, whereas the objective response rate was 70.3%. Our results showed that nivolumab plus chemotherapy is effective and safe for first-line treatment of Turkish patients with HER2-negative advanced gastric, GEJ, or esophageal adenocarcinoma. Patient selection is crucial for optimal outcomes. Future studies are needed to identify predictive biomarkers and treatment strategies to further improve the prognosis of patients.

**Abstract:**

Based on the CheckMate 649 trial, nivolumab plus chemotherapy is the recommended first-line treatment for HER2-negative unresectable advanced or metastatic gastric, gastroesophageal junction (GEJ), or esophageal adenocarcinoma. This nationwide, multicenter, retrospective study evaluated the real-world effectiveness of this regimen in Turkish patients and identified subgroups that may experience superior outcomes. Conducted across 16 oncology centers in Turkey, this study retrospectively reviewed the clinical charts of adult patients diagnosed with HER2-negative unresectable advanced or metastatic gastric, GEJ, or esophageal adenocarcinoma from 2016 to 2023. This study included 111 patients (54 women, 57 men) with a median age of 58 years. The median progression-free survival (PFS) and overall survival (OS) were 11.7 months and 18.2 months, respectively, whereas the objective response rate (ORR) was 70.3%. Multivariable analyses revealed that previous curative surgery was a favorable independent prognostic factor for both PFS and OS. Conversely, an Eastern Cooperative Oncology Group performance status of 2 emerged as an adverse independent prognostic factor for OS. The safety profile of nivolumab plus chemotherapy was found to be manageable. Our findings support the use of nivolumab plus chemotherapy for the first-line treatment of Turkish patients with HER2-negative unresectable advanced or metastatic gastric, GEJ, or esophageal adenocarcinoma. Patient selection based on clinical characteristics is crucial for optimizing treatment outcomes.

## 1. Introduction

Gastric, gastroesophageal junction (GEJ), and esophageal cancers are among the most prevalent and lethal malignancies worldwide [1,2]. Adenocarcinomas constitute the vast majority (>90%) of gastric and GEJ cancers and a significant proportion of esophageal malignancies, particularly in Western countries [3,4,5]. Until recently, the standard first-line treatment for patients with unresectable advanced or metastatic human epidermal growth factor receptor 2 (HER2)-negative gastric, GEJ, and esophageal adenocarcinomas consisted of fluoropyrimidine plus platinum-based chemotherapy [6,7,8]. Unfortunately, this approach provided only limited efficacy, with a median overall survival (OS) of approximately 11–12 months [9,10,11].

The introduction of immune checkpoint inhibitors has recently transformed the treatment landscape, leading to a paradigm shift for first-line treatment of patients with programmed death-ligand 1 (PD-L1)-positive, advanced or metastatic, non-HER2-positive gastric, GEJ, and esophageal adenocarcinomas [12,13,14]. The CheckMate 649 trial, a phase 3, randomized, open-label study, showcased enhanced outcomes when nivolumab—a fully human IgG4 monoclonal antibody that targets PD-1—was combined with chemotherapy as a first-line treatment for patients with tumors expressing PD-L1 and a combined positive score (CPS) of 5 or higher [15]. The addition of nivolumab to chemotherapy resulted in a median OS of 14.4 months, compared to 11.1 months with chemotherapy alone [15]. Similarly, the median progression-free survival (PFS) improved to 7.7 months from 6.1 months with combination therapy versus chemotherapy alone, respectively [15]. Moreover, patients receiving the nivolumab and chemotherapy combination demonstrated a superior objective response rate (ORR), achieving more sustained responses and a higher incidence of complete responses than those treated with chemotherapy alone [15]. Given the absence of new safety signals, the U.S. Food and Drug Administration (FDA) approved nivolumab in combination with chemotherapy in April 2021 as the first-line treatment for metastatic gastric cancer and esophageal adenocarcinoma [16,17]. Notably, in a 3-year follow-up study of the CheckMate 649 trial, the addition of nivolumab to chemotherapy maintained clinically meaningful long-term survival benefits compared to chemotherapy alone, with an acceptable safety profile [18].

Real-world observational studies have increasingly proven to be a valuable tool for evaluating the benefits and limitations of guideline-recommended therapies in more representative patient populations [19,20]. These investigations provide crucial insights into treatment outcomes in less controlled clinical settings [21,22]. To further elucidate the real-world effectiveness of first-line nivolumab plus chemotherapy in Turkish patients with HER2-negative unresectable advanced or metastatic gastric, GEJ, or esophageal adenocarcinoma, we have designed a nationwide, multicenter observational study. We also sought to identify specific patient subgroups that may experience superior outcomes from this treatment regimen. The insights derived from the current study will not only supplement the findings from randomized controlled trials but also facilitate the development of a more individualized approach to implementing this combination therapy in clinical practice.

## 2. Methods

### 2.1. Study Design and Participants

This nationwide, multicenter, non-interventional, retrospective study reviewed the medical records of adult patients diagnosed with HER2-negative unresectable advanced or metastatic gastric, GEJ, or esophageal adenocarcinoma. This study was conducted across 16 oncology centers in Turkey and examined clinical charts from 2021 to 2023. All participants were 18 years of age or older. In line with the treatment protocol established in the CheckMate 649 clinical trial [15], the patients received a combination therapy consisting of standard chemotherapy regimens and nivolumab. Patients were administered nivolumab 360 mg every 3 weeks or 240 mg every 2 weeks. Two distinct chemotherapy regimens were employed: FOLFOX and XELOX. The FOLFOX regimen was administered intravenously over a 48 h period every 2 weeks (85 mg/m^2^ oxaliplatin day 1, 400 mg/m^2^ leucovorin day 1, 400 mg/m^2^ 5-fluorouracil day 1, and 2400 mg/m^2^ 5-fluorouracil 48 h via continuous infusion). In the XELOX regimen, capecitabine (1000 mg/m^2^, days 1–14) was administered orally twice daily for two weeks, followed by a one-week rest period, and oxaliplatin (130 mg/m^2^, day 1) was given intravenously on the first day of each three-week cycle.

Patient staging was conducted in accordance with the Eighth Edition of the *American Joint Committee on Cancer/Union for International Cancer Control Staging Manual*, utilizing both clinical and radiological findings. All patients included in this study presented with either unresectable advanced or metastatic disease at the time of diagnosis. To be eligible for inclusion, patients were required to exhibit an Eastern Cooperative Oncology Group (ECOG) performance status of 0 (fully active, able to carry out pre-disease activities without restriction), 1 (restricted in strenuous activities but ambulatory and able to perform light work), or 2 (ambulatory and capable of all self-care but unable to carry out any work activities; up and about more than 50% of waking hours) [23]. The study protocol was reviewed and approved (reference number: E-10840098-772.02-5812) by the Ethics Committee at Medipol University (Istanbul, Turkey). Written informed consent was obtained from all patients or their designated legal representatives.

### 2.2. Data Collection

The following patient data were gathered from clinical records: age, sex, ECOG performance status, primary tumor location, history of prior curative surgery, initial disease stage, presence of Signet ring cell carcinoma, number and locations of metastases, and the chemotherapy regimen administered (either FOLFOX or XELOX). Additionally, the PD-L1 CPS was determined by dividing the total count of PD-L1-stained tumor and immune cells by the total number of viable tumor cells, then multiplying by 100, yielding a maximum score of 100 [24]. The PD-L1 CPS data were analyzed using clinically relevant cut-offs of ≥1, ≥5, and ≥10 [24]. Tumor cell PD-L1 expression and PD-L1 CPS were evaluated using the Dako PD-L1 IHC 28-8 pharmDx assay (Dako, an Agilent Technologies Inc. company, Santa Clara, CA, USA). Tumor cell PD-L1 expression was defined as the percentage of viable tumor cells with partial or complete membrane staining in at least 100 viable tumor cells. CPS was generated by re-scoring PD-L1-stained slides and was defined as the number of PD-L1-positive tumor cells with partial or complete membrane staining, plus lymphocytes and macrophages with membrane staining, intracellular staining or both, divided by the total viable tumor cells multiplied by 100.

### 2.3. Efficacy Measures

The primary efficacy measures included PFS, OS, and the ORR. PFS was defined as the time elapsed from the initiation of treatment until objective tumor progression, death from any cause, or the date of the last follow-up evaluation, whichever occurred first. OS was calculated as the duration from the start of treatment until the patient’s death, irrespective of the underlying cause, or until the date of the last follow-up assessment if the patient remained alive. The evaluation of treatment response followed the Response Evaluation Criteria in Solid Tumors (RECIST) guidelines, version 1.1 [25]. Specifically, responses were classified into four categories: complete response (CR), partial response (PR), progressive disease (PD), and stable disease (SD) [25]. The ORR quantified the percentage of patients who attained either a complete or partial response. Furthermore, the disease control rate (DCR) was computed to include patients who achieved complete response, partial response, or maintained stable disease [25].

### 2.4. Safety Endpoints

Treatment-related adverse events (TRAEs) were graded in accordance with the National Cancer Institute Common Terminology Criteria for Adverse Events, version 5.0 [26]. The incidence of TRAEs was reported separately for grade 1–2 (mild to moderate) and grade 3–4 (severe to life-threatening) events [26]. For the purposes of analysis, TRAEs of special interest included nausea, vomiting, peripheral neuropathy, diarrhea, fatigue, weight loss, decreased appetite, stomatitis, elevated lipase levels, hypothyroidism, neutropenia, elevated alanine aminotransferase and aspartate aminotransferase levels, rash, alopecia, anemia, and thrombocytopenia. The proportion of patients experiencing each TRAE was reported for the overall study population.

### 2.5. Data Analysis

Data from all participating centers were pooled for analysis. Variables are expressed using descriptive statistics (counts, percentages, means, standard deviations, medians, and ranges). To visualize survival estimates, we generated Kaplan–Meier plots and used the log-rank test for statistical comparison. To analyze the relationships between the variables under study and survival outcomes, we conducted both univariable and multivariable Cox proportional hazards regression analyses. We adopted a stepwise selection approach, incorporating significant variables from the univariable analysis into the multivariable model. The results are presented as hazard ratios (HRs) with their corresponding 95% confidence intervals (CIs). Analyses were conducted using the SPSS software, version 24.0 (IBM, Armonk, NY, USA). Two-tailed *p* values ˂ 0.05 were considered statistically significant.

## 3. Results

### 3.1. Patient Characteristics

Table 1 presents the general characteristics of the 111 study patients (54 women and 57 men; median age: 58 years). The primary tumor locations were predominantly gastric adenocarcinoma (88 patients, 79.2%), followed by GOJ adenocarcinoma (21 patients, 18.9%) and esophageal adenocarcinoma (2 patients, 1.9%). A minority of the study participants (24.3%) had undergone previous curative surgery, whereas 75.7% had not. Regarding the initial disease stage, 24 patients (21.6%) were diagnosed with locally advanced cancer, whereas 87 (78.4%) presented with metastatic disease. The liver (43.2%) and peritoneum (39.6%) were the most common metastatic sites. The most commonly prescribed chemotherapy regimen was FOLFOX, administered to a substantial majority of 107 patients (96.4%). In contrast, the XELOX regimen was used in four patients only (3.6%). Regarding the expression of the PD-L1 biomarker, the vast majority of patients, 106 (95.5%), exhibited a CPS greater than 1. Furthermore, a substantial proportion of 100 patients (90.1%) had a CPS exceeding 5, and 76 patients (68.5%) demonstrated a CPS higher than 10, suggesting elevated levels of PD-L1 expression in a significant subset of the cohort.

### 3.2. Survival Outcomes

The study cohort exhibited a median PFS of 11.7 months (95% CI = 10.2–13.2 months; Figure 1). Table 2 presents the results of univariable and multivariable analyses, which aimed to identify predictors of PFS within the study cohort. After adjusting for potential confounding factors, the multivariable analysis revealed that previous curative surgery was the sole independent predictor associated with a more favorable PFS (HR = 0.33, 95% CI = 0.13−0.85, *p* = 0.022). Regarding OS, the study participants demonstrated a median of 18.2 months (95% CI = 15.0−21.2 months; Figure 2). Table 3 summarizes the findings of univariable and multivariable analyses conducted to identify predictors of OS. After accounting for potential confounders, the multivariable analysis indicated that previous curative surgery was the only independent predictor associated with a more favorable OS (HR = 0.52, 95% CI = 0.16−0.62, *p* = 0.026). Conversely, an ECOG performance status of 2 emerged as an adverse independent prognostic factor (HR = 3.34, 95% CI = 1.20−9.31, *p* = 0.021).

### 3.3. Treatment Response

Of the 111 patients, 12 (10.8%) achieved CR, 66 (59.5%) had PR, 18 (16.2%) maintained SD, and 15 (13.5%) experienced PD. The ORR, defined as the percentage of patients attaining either CR or PR, was 70.3%. The DCR, which includes patients with CR, PR, or SD, was 86.5% (Table 4). The median number of nivolumab cycles was 11 (range: 6–35), while the median number of chemotherapy cycles was 12 (range: 6–31).

### 3.4. Treatment-Related Adverse Events

Table 5 summarizes the most commonly reported TRAEs during chemotherapy plus nivolumab treatment. The majority were grade 1 or 2 in severity. The most frequent grade 1−2 events included nausea (52.2%), anemia (43.2%), fatigue (37.8%), peripheral neuropathy (28.8%), diarrhea (28.8%), neutropenia (28.8%), and stomatitis (26.1%). Grade 3 or 4 AEs were less common, with the most prevalent being anemia (9.9%), neutropenia (7.2%), thrombocytopenia (4.5%), and fatigue (4.5%). Notably, no grade 3–4 events were reported for weight loss, stomatitis, lipase increase, rash, or alopecia.

## 4. Discussion

Based on the findings from the CheckMate 649 clinical trial [15], the European regulatory authorities approved nivolumab in combination with chemotherapy for the first-line treatment of HER2-negative unresectable advanced or metastatic gastric, GEJ, or esophageal adenocarcinoma in patients with a PD-L1 CPS ≥ 5 [16,17]. In contrast, regulatory bodies in the United States and Asia granted approval for this combination therapy in the same patient population, irrespective of PD-L1 CPS [16,17]. To evaluate the real-world efficacy and safety of this combination therapy in Turkey, a nationwide, multicenter observational study was conducted, generating evidence on the treatment outcomes in this specific patient group.

Our investigation yielded three principal findings. First, in terms of efficacy, we observed a median PFS of 11.7 months and a median OS of 18.2 months. In addition, the ORR in the entire study cohort was 70.3%. These results compare favorably with those reported in the 3-year follow-up of the CheckMate 649 trial for patients whose tumors expressed PD-L1 CPS ≥ 5, where Janjigian et al. [18] reported a median OS of 14.4 months, a median PFS of 8.3 months, and an ORR of 60%. Similarly, in the CheckMate 649 Chinese subgroup analysis, Liu et al. [27] reported a median OS of 14.3 months, a median PFS of 8.3 months, and an ORR of 66%. While our real-world outcomes appear promising in the context of the available evidence from controlled clinical trial settings, it is important to note that direct comparisons should be interpreted with caution due to potential differences in patient ethnicity, follow-up durations, study design, and other confounding factors. Second, our analysis identified a history of previous curative surgery as a favorable prognostic factor for both PFS and OS, whereas an ECOG performance status of 2 emerged as an adverse independent prognostic factor for OS. Several potential explanations exist for why a history of previous curative surgery may be a favorable prognostic factor in patients who received nivolumab plus chemotherapy. Prior curative surgery may indicate that this subgroup had an earlier stage, more localized disease that was amenable to resection before progressing to the advanced/metastatic setting, suggesting they may have had less aggressive tumor biology compared to patients who presented with unresectable disease from the start [28]. Moreover, patients who previously underwent curative-intent surgery are likely to have a lower prevalence of comorbidities. In addition, resection of the primary tumor reduces the overall tumor burden, even if the malignancy subsequently recurs or progresses. The immune system may be better able to mount an anti-tumor response facilitated by nivolumab when disease volume is lower after prior resection [28]. In contrast, the negative impact of an ECOG performance status of 2 likely reflected increased tumor-related morbidity and reduced ability to tolerate cancer-directed therapies [23]. This finding underscores the importance of assessing functional status in both clinical trial design and real-world management of advanced gastroesophageal cancer. The safety profile of nivolumab plus chemotherapy in our real-world cohort was found to be manageable, providing reassuring evidence that the toxicity profile of this combination in routine clinical practice largely mirrors what has been established in more controlled clinical trial settings. The most common grade 1–2 adverse events observed in our study, such as nausea, fatigue, peripheral neuropathy, and diarrhea, were also among the most frequent low-grade toxicities reported in clinical trials [15,18,27]. In addition, the incidence of more severe grade 3–4 adverse events was relatively low in the current research, aligning with clinical trial data [15,18,27,29]. Interestingly, immune-related adverse events, including hypothyroidism and rash, occurred at low rates (6.3% each), which is also consistent with reports from checkpoint inhibitor studies in clinical trial settings. However, the rates of nausea, vomiting, and stomatitis appear to be higher in our real-world population compared to trial cohorts [15,18,27]. This observation may be explained by several factors. In real-world settings, patients frequently present with a higher burden of comorbidities and poorer overall health status compared to the carefully selected participants enrolled in clinical trials. Furthermore, concomitant medications that real-world patients may be taking for other medical conditions could interact with the checkpoint inhibitors, thereby exacerbating gastrointestinal adverse effects. Lastly, real-world patient populations likely include a higher proportion of frail individuals who are at an elevated risk for treatment-related toxicities but are frequently underrepresented in clinical trial cohorts.

Our findings should be interpreted in the context of several limitations. First, the retrospective nature of the analysis may introduce potential biases and confounding factors that could impact the interpretation of the results. Prospective studies with predefined data collection and analysis plans would provide more robust evidence to support the conclusions drawn from this study. Second, the small number of patients with esophageal adenocarcinoma included in the analysis limits the generalizability of the findings to this specific subgroup. Esophageal adenocarcinoma may have distinct biological and clinical characteristics compared to gastric and GEJ adenocarcinomas [3], and the limited sample size may not adequately capture the heterogeneity of treatment responses in this population. Future studies with larger cohorts of patients with esophageal adenocarcinoma are needed to validate the efficacy and safety of nivolumab plus chemotherapy in this specific subtype. Third, the limited number of patients who received the XELOX chemotherapy regimen in combination with nivolumab may not provide a comprehensive assessment of the efficacy and safety profile of this particular treatment combination. Different chemotherapy regimens may have varying degrees of synergistic or additive effects when combined with nivolumab. Fourth, this study did not analyze patient-reported outcomes [1] or quality-adjusted survival [30], which are important considerations in the evaluation of cancer treatments. Since our study is a study containing real-life data, it does not include a control arm like other similar real-life studies in the literature. In addition, the nivolumab chemotherapy combination is now the standard of care treatment for patients with HER2-negative unresectable advanced or metastatic gastric, GEJ, or esophageal adenocarcinoma with PD-L1 CPS ≥ 5; thus, patients with these characteristics should receive this treatment as long as it is available. Incorporating these measures in future studies would provide a more comprehensive assessment of the benefits and risks of nivolumab plus chemotherapy in this patient population.

## 5. Conclusions

In conclusion, this real-world study demonstrates the efficacy and safety of nivolumab in combination with chemotherapy for the first-line treatment of Turkish patients with HER2-negative unresectable advanced or metastatic gastric, GEJ, or esophageal adenocarcinoma. Our findings support the use of this combination therapy in clinical practice and highlight the importance of patient selection based on clinical characteristics to optimize treatment outcomes. Ongoing research efforts should focus on identifying predictive biomarkers and refining treatment strategies to further improve the prognosis of patients with these challenging malignancies.

## Figures and Tables

**Figure 1 cancers-16-02251-f001:**
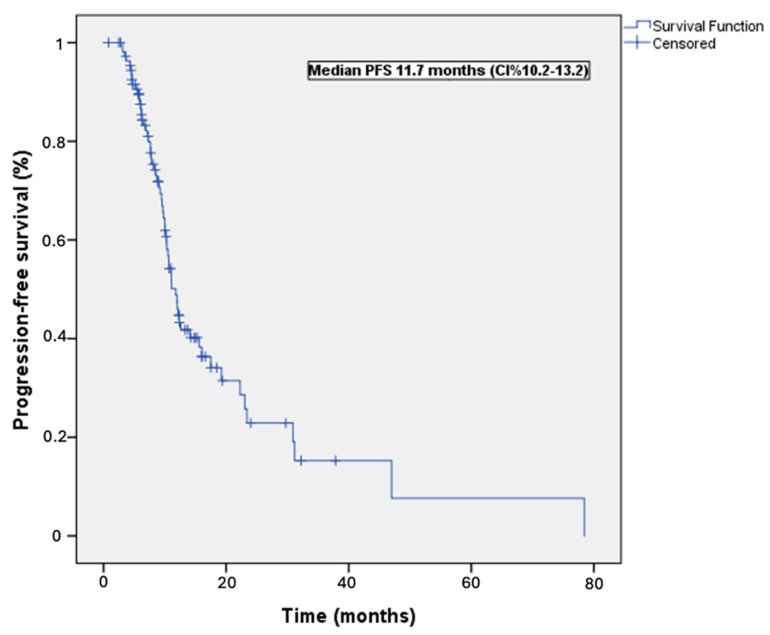
Progression-free survival curve, Kaplan–Meier plot of progression-free survival in the 111 study patients.

**Figure 2 cancers-16-02251-f002:**
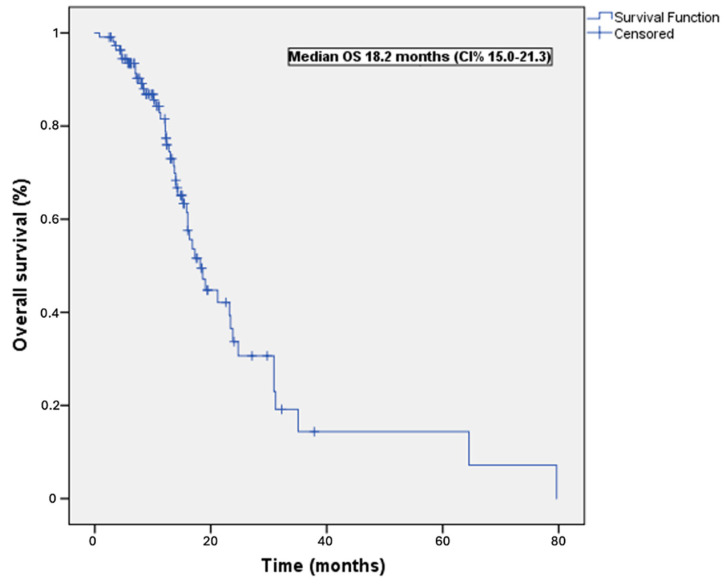
Overall survival curve. Kaplan–Meier plot of overall survival in the 111 study patients.

**Table 1 cancers-16-02251-t001:** General characteristics of the 111 study patients.

Characteristic	n (%)
Sex	
Female	54 (48.6)
Male	57 (51.4)
Median age, years	58 (31–82)
<65	79 (71.2)
>65	32 (28.8)
ECOG performance status	
0	50 (45.1)
1	49 (44.1)
2	12 (10.8)
Primary tumor location	
Gastric adenocarcinoma	88 (79.2)
GOJ adenocarcinoma	21 (18.9)
Oesophageal adenocarcinoma	2 (1.9)
Previous curative surgery	
Present	27 (24.3)
Absent	84 (75.7)
Initial disease stage	
Locally advanced	24 (21.6)
Metastatic	87 (78.4)
Signet ring cell carcinoma	
Present	22 (19.9)
Absent	89 (80.1)
Number of metastatic sites	
1	59 (53.2)
≥2	52 (46.8)
Site of metastasis	
Liver	48 (43.2)
Lung	19 (17.1)
Peritoneum	44 (39.6)
Bone	12 (10.8)
CNS	1 (0.9)
Distant lymph nodes	35 (31.5)
Other sites	4 (3.6)
Chemotherapy regimen	
FOLFOX	107 (96.4)
XELOX	4 (3.6)
PD-L1 CPS	
<1	5 (4.5)
≥1	106 (95.5)
≥5	100 (90.1)
≥10	76 (68.5)

Abbreviations: ECOG, Eastern Cooperative Oncology Group; GOJ, gastroesophageal junction; CNS, central nervous system; FOLFOX, folinic acid, fluorouracil, oxaliplatin; XELOX, capecitabine plus oxaliplatin; PD-L1: programmed death-ligand 1; CPS, combined positive score.

**Table 2 cancers-16-02251-t002:** Univariable and multivariable analysis of progression-free survival.

Characteristic	n (%)	Median PFS (Months)	Univariable *p* Value	Multivariable *p* Value	HR (95 CI%)
Sex					
Female	54 (48.6)	10.6	0.07		
Male	57 (51.4)	12.1			
Median age (interquartile range), years	58 (31–82)				
<65	79 (71.2)	11.0	0.75		
>65	32 (28.8)	12.0			
ECOG performance status					
0	50 (45.1)	12.0	0.038		
1	49 (44.1)	12.1			
2	12 (10.8)	10.3		0.26	1.57 (0.70–3.54)
Primary tumor location					
Gastric adenocarcinoma	88 (79.2)	11.5	0.44		
GOJ adenocarcinoma	21 (18.9)	12.1			
Esophageal adenocarcinoma	2 (1.9)	11.4			
Previous curative surgery					
Present	27 (24.3)	25.7	<0.001	0.022	0.33 (0.13–0.85)
Absent	84 (75.7)	10.0			
Initial disease stage					
Locally advanced	24 (21.6)	23.0	<0.001	0.12	2.00 (0.81–4.90)
Metastatic	87 (78.4)	10.1			
Signet ring cell carcinoma					
Present	22 (19.9)	10.1	0.82		
Absent	89 (80.1)	12.3			
Number of metastatic sites					
1	59 (53.2)	12.0	0.22		
≥2	52 (46.8)	11.0			
Site of metastasis					
Liver	48 (43.2)	10.6	0.65		
Lung	19 (17.1)	11.1			
Peritoneum	44 (39.6)	12.3			
Bone	12 (10.8)	10.5			
CNS	1 (0.9)	NA			
Distant lymph nodes	35 (31.5)	18.2			
Other sites	4 (3.6)	8.7			
Chemotherapy regimen					
FOLFOX	107 (96.4)	11.1	0.96		
XELOX	4 (3.6)	13.5			
PD-L1 CPS					
<5	20 (18.1)	11.1	0.031	0.35	1.62 (0.58–4.48)
≥5	91 (81.9)	17.5			

Abbreviations: PFS, progression-free survival; HR, hazard ratio; CI, confidence interval; ECOG, Eastern Cooperative Oncology Group; GOJ, gastroesophageal junction; CNS, central nervous system; NA, not applicable; FOLFOX, folinic acid, fluorouracil, oxaliplatin; XELOX, capecitabine plus oxaliplatin; PD-L1: programmed death-ligand 1; CPS, combined positive score.

**Table 3 cancers-16-02251-t003:** Univariable and multivariable analysis of overall survival.

Characteristic	n (%)	Median OS (Months)	Univariable *p* Value	Multivariable *p* Value	HR (95 CI%)
Sex					
Female	54 (48.6)	17.2	0.19		
Male	57 (51.4)	19.0			
Median age (interquartile range), years	58 (31–82)				
<65	79 (71.2)	17.2	0.20		
>65	32 (28.8)	23.2			
ECOG performance status					
0	50 (45.1)	21.2	0.032		
1	49 (44.1)	16.0			
2	12 (10.8)	11.3		0.021	3.34 (1.20–9.31)
Primary tumor location					
Gastric adenocarcinoma	88 (79.2)	18.1	0.45		
GOJ adenocarcinoma	21 (18.9)	17.2			
Esophageal adenocarcinoma	2 (1.9)	16.8			
Previous curative surgery					
Present	27 (24.3)	16.0	0.005	0.026	0.52 (0.16–0.62)
Absent	84 (75.7)	23.8			
Initial disease stage					
Locally advanced	24 (21.6)	23.4	0.025	0.66	1.27 (0.42–3.82)
Metastatic	87 (78.4)	16.0			
Signet ring cell carcinoma					
Present	22 (19.9)	18.2	0.25		
Absent	89 (80.1)	16.8			
Number of metastatic sites					
1	59 (53.2)	21.2	0.22		
≥2	52 (46.8)	16.8			
Site of metastasis					
Liver	48 (43.2)	23.2	0.39		
Lung	19 (17.1)	13.7			
Peritoneum	44 (39.6)	18.2			
Bone	12 (10.8)	19.0			
CNS	1 (0.9)	NA			
Distant lymph nodes	35 (31.5)	19.7			
Other sites	4 (3.6)	NA			
Chemotherapy regimen					
FOLFOX	107 (96.4)	18.2	0.83		
XELOX	4 (3.6)	23.4			
PD-L1 CPS					
<5	20 (18.1)	17.2	0.29	0.24	2.34 (0.55–9.85)
≥5	91 (81.9)	18.6			

Abbreviations: OS, overall survival; HR, hazard ratio; CI, confidence interval; ECOG, Eastern Cooperative Oncology Group; GOJ, gastroesophageal junction; CNS, central nervous system; NA, not applicable; FOLFOX, folinic acid, fluorouracil, oxaliplatin; XELOX, capecitabine plus oxaliplatin; PD-L1: programmed death-ligand 1; CPS, combined positive score.

**Table 4 cancers-16-02251-t004:** Patterns of treatment response observed in this study.

Response	n	%
Complete response	12	10.8
Partial response	66	59.5
Stable disease	18	16.2
Progressive disease	15	13.5
Objective response rate	78	70.3
Disease control rate	96	86.5

**Table 5 cancers-16-02251-t005:** Treatment-related adverse events observed in this study.

Adverse Event	Grade 1 or 2, n (%)	Grade 3 or 4, n (%)
Nausea	58 (52.2)	2 (1.8)
Vomiting	19 (6.8)	2 (1.8)
Peripheral neuropathy	32 (28.8)	3 (2.7)
Diarrhea	32 (28.8)	1 (0.9)
Fatigue	42 (37.8)	5 (4.5)
Weight loss	11 (9.9)	-
Decreased appetite	14 (12.6)	1 (0.9)
Stomatitis	29 (26.1)	2 (1.8)
Lipase increased	7 (6.3)	-
Hypothyroidism	7 (6.3)	2 (1.8)
Neutropenia	32 (28.8)	8 (7.2)
Increased alanine aminotransferase	18 (16.2)	1 (0.9)
Increased aspartate aminotransferase	21 (18.9)	1 (0.9)
Rash	7 (6.3)	-
Alopecia	12 (10.8)	-
Anemia	48 (43.2)	11 (9.9)
Thrombocytopenia	20 (18.0)	5 (4.5)

## Data Availability

Data will be available from the corresponding author upon reasonable request.

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
