# Peer review of "Real-World Efficacy and Safety of First-Line Nivolumab Plus Chemotherapy in Patients with Advanced Gastric, Gastroesophageal Junction, and Esophageal Adenocarcinoma: A Nationwide Observational Turkish Oncology Group (TOG) Study"

_cancers, 2024, doi:10.3390/cancers16122251_

Round 1

Reviewer 1 Report

Comments and Suggestions for Authors

1. Need to write the conclusions section. .Currently, it looks unattended to.

2. You mentioned reasons for previous curative surgery having significantly better overall survival. Therefore, please mention that this finding might be correlative rather than suggestive of better outcomes with nivolumab. Please avoid any mention of the applicability of using this  parameter for patient selection (as suggested in the last paragraph in discussion)

3.  Similarly with ECOG 2 score, please avoid mention of this as a significant clinical parameter for patient selection for nivolumab. These retrospective studies are very correlative as suggested by you since patients with lower tumor burden will generally have better outcomes. 

4. How did you estimate efficacy without having a control group?

5. Descriptive statistics presented are interesting but no comments could be made about efficacy without a control group in your retrospective observational study. 

Author Response

Thank you for your comments. 

  1. The conclusions section has been written as a separate heading as suggested.
  2. The mention of using previous curative surgery as a parameter for patient selection has been removed from the discussion section.
  3. The mention of ECOG 2 score as a significant clinical parameter for patient selection for nivolumab has been removed from the discussion section.
  4. and 5. We have added the following explanation to the discussion section: "Since our study is a study containing real-life data, it does not include a control arm like other similar real-life studies in the literature. In addition, nivolumab chemotherapy combination is now the standard of care treatment for patients with HER2-negative unresectable advanced or metastatic gastric, GEJ, or esophageal adenocarcinoma with PD-L1 CPS ≥5, thus, patients with these characteristics should receive this treatment as long as it is available."

Additionally, the methods and results sections have been expanded and improved.

Reviewer 2 Report

Comments and Suggestions for Authors

This manuscript analyzed the effectiveness and safety of the combined treatment of nivolumab and chemotherapies for HER2-negative unresectable advanced or metastatic gastric, gastroesophageal geal junction(GEJ), or esophageal adenocarcinoma in the Turkish patients. This study is well designed and organized, and the reported results are interesting to readers. The only concern is that the conclusion paragraph should be moved under the “Conclusion” subtitle.

Comments on the Quality of English Language

I would suggest revising the manuscript's English languages due to some grammar errors.

Author Response

Thank you for your comments.

The conclusions section has been written as a separate heading as suggested.

The manuscript has been thoroughly revised for grammar and language errors.

Reviewer 3 Report

Comments and Suggestions for Authors

It is well written and clear. It is a straithforward description of the outcomes of the cohort. There is nothing that needs to be done. There is a few spaces missing in the abstract and the heading for the conclusion is not in the right place.

Author Response

Thank you for your positive feedback and suggestions. We have corrected the missing spaces in the abstract and repositioned the heading for the conclusion as recommended.

Round 2

Reviewer 1 Report

Comments and Suggestions for Authors

Thank  you for the changes.